# Indoleamine 2,3-Dioxygenase Regulates Placental Trophoblast Cell Invasion

**DOI:** 10.3390/ijms26125889

**Published:** 2025-06-19

**Authors:** Yoshiki Kudo, Jun Sugimoto

**Affiliations:** Department of Obstetrics and Gynecology, Graduate School of Biomedical Sciences, Hiroshima University, Hiroshima 734-8551, Japan; yoshkudo@hiroshima-u.ac.jp

**Keywords:** trophoblast invasion, indoleamine 2,3-dioxygenase (IDO), transwell assay, HTR-8/SVneo cell, decidua, placenta accreta spectrum

## Abstract

To clarify the physiological importance of the tryptophan catabolizing enzyme, indoleamine 2,3-dioxygenase, in human pregnancy, we have studied how the expression of this enzyme controls extravillous cytotrophoblast invasion into the decidua. We have generated an Ishikawa cell line stably transfected with a plasmid encoding indoleamine 2,3-dioxygenase under the control of a tetracycline inducible promoter. Using this Ishikawa cell line and extravillous cytotrophoblast cell line, HTR-8/SVneo, we developed a quantitative in vitro trophoblast invasion assay. When trophoblast cells were cultured on a layer of Ishikawa cells expressing indoleamine 2,3-dioxygenase, tryptophan degradation was enhanced and trophoblast cell invasion was suppressed. These findings suggest that indoleamine 2,3-dioxygenase expressed in the decidua may play a role in regulating trophoblast invasion.

## 1. Introduction

During implantation, cytotrophoblast cells in the chorionic villi come into contact with the maternal decidua, differentiate into extravillous cytotrophoblasts, and invade the maternal decidua as interstitial extravillous cytotrophoblasts. This process of trophoblast invasion into maternal tissue is considered regulated by mechanisms within the decidual layer [1,2]. The regulatory mechanisms determining the extent of extravillous cytotrophoblast invasion remain poorly understood [3]. Immune cells, including macrophages and uterine natural killer cells, infiltrate the decidua and are thought to play a role in regulating trophoblast invasion. In vitro studies have shown that indoleamine 2,3-dioxygenase (IDO) expressed by macrophages can actively induce apoptosis in extravillous cytotrophoblast cells through tryptophan depletion mediated by IDO [4]. Additionally, this study suggested that, in vivo, IDO-mediated tryptophan depletion by decidual macrophages may similarly contribute to the induction of apoptosis and the regulation of trophoblast invasion [4].

IDO, an enzyme widely expressed across various mammalian tissues, serves as the first and rate-limiting enzyme in the tryptophan metabolism thorough the kynurenine pathway [5]. IDO has been implicated not only in reproductive biology [6] but also in several pathophysiological conditions of clinical significance, including inflammatory diseases, autoimmune disorders, infections, neuropathology, cancer, and organ transplantation [7,8,9,10]. The human placenta is a tissue with particularly high IDO activity [11]. Our study demonstrated a robust expression of IDO in the glandular epithelium and macrophages of the first trimester decidua [12]. Similarly, Sedlmayer et al. reported strong IDO expression in the glandular epithelium, with some positive cells also observed in the decidual stroma [13].

Placental implantation at the site of a previous cesarean scar represents a serious pregnancy complication characterized by trophoblast invasion into the myometrial layer, leading to placenta accreta and percreta (placenta accreta spectrum). We assessed the expression of IDO in a sample from a cesarean scar pregnancy, which was obtained from a hysterectomy performed due to placental implantation at the site of a prior cesarean section scar [14]. IDO expression was clearly observed in the glandular epithelium and macrophages of the decidua. Extravillous cytotrophoblast invasion was detected in the myometrium areas not covered by decidual tissue, while trophoblast invasion into the myometrium was absent in regions where this tissue was covered by decidual tissue. Based on these findings, we hypothesize that the expression of IDO in the maternal decidua may be involved in the regulation of trophoblast invasion, and that its absence in areas lacking decidual tissue may contribute to the pathogenesis of the over-invaded placenta [1].

In this study, we developed an in vitro invasion assay using trophoblast cells (HTR-8/SVneo cells) and Ishikawa cells stably transfected with a plasmid encoding IDO under the control of a tetracycline inducible promoter. We then conducted experiments in order to further explore the potential role of IDO in regulating trophoblast invasion.

## 2. Results

### 2.1. Ishikawa Cells Expressing IDO

Tet-On Ishikawa cells were transfected with plasmid encoding IDO, and several lines stably expressing IDO protein were obtained; one line highly expressing IDO protein was chosen for further analysis (Ishikawa Tet-On IDO cell), and the results obtained with this line are described.

Figure 1A shows a Western blot analysis of IDO protein levels in extracts of control cells and of Ishikawa Tet-On IDO cells followed by 22 h of doxycycline or vehicle treatment. The expected band, corresponding to the predicted molecular weight of IDO, was found at 45 kDa. In Ishikawa Tet-On IDO cells, doxycycline treatment produced a marked stimulation in IDO protein level compared with control cells or with vehicle. Different concentrations of doxycycline over the range of 50 to 2000 ng/mL stimulated the IDO protein level in a concentration-dependent manner (Figure 1B). Significant and maximal stimulation were observed at 50 and 1000 ng/mL of doxycycline, respectively (Figure 1C).

Functional IDO activity was studied by measuring concentrations of tryptophan and its degradation product kynurenine in the conditioned medium (Figure 2). Both tryptophan degradation and kynurenine appearance correlated well with the level of IDO protein determined by Western blot (Figure 1B,C). Hence, the ratios of the IDO product (kynurenine) to substrate (tryptophan), an index of tryptophan catabolism, were significantly increased in a doxycycline concentration-dependent manner (Figure 2C).

### 2.2. An Assay for HTR-8/SVneo Cell Invasion

We next developed a quantitative assay of cell invasion. HTR-8/SVneo cells stably expressing GFP (HTR8 GFP cell) were cultured on a layer of Ishikawa Tet-On IDO cells in the absence or presence of doxycycline. Then, the number of HTR8 GFP cells which appeared on the opposite side of the permeable membrane was assessed (Figure 3A). The number of cells expressing GFP, that invaded HTR8 GFP cells, was clearly decreased following IDO expression induced by doxycycline treatment (Figure 3B). As predicted, both the decrease in tryptophan concentration (vehicle, 81 ± 6; doxycycline, 74 ± 3 μmol/L) and the associated increase in kynurenine concentration (vehicle, 0.81 ± 0.10; doxycycline, 6.26 ± 1.55 μmol/L) in the conditioned medium were enhanced significantly by the presence of 200 ng/mL of doxycycline in the culture medium.

## 3. Discussion

The experiments described here were conducted to investigate the functional role of IDO in regulating trophoblast cell invasion. For experimental purposes, we used the model human trophoblast cell line HTR-8/SVneo, which is known to exhibit many of the characteristics of freshly isolated human trophoblasts [15,16]. The Ishikawa cell line, derived from a uterine endometrial adenocarcinoma [17], was employed as a surrogate for primary uterine endometrial cells. We found experimental evidence that IDO-mediated tryptophan degradation regulates trophoblast invasion. This conclusion arises from the following observations: (1) doxycycline treatment produced a marked stimulation in IDO protein level in Ishikawa cells stably transfected with a plasmid encoding IDO under the control of a tetracycline inducible promoter (Ishikawa Tet-On IDO) (Figure 1A); (2) the functional IDO activity determined by measuring concentrations of tryptophan and kynurenine in the conditioned medium of Ishikawa Tet-On IDO cells correlated well with IDO protein level (Figure 1B,C and Figure 2); (3) the invasion of HTR-8/SVneo trophoblast cells evaluated using the newly developed invasion assay was suppressed by the expression of IDO in Ishikawa Tet-On IDO cells following doxycycline treatment (Figure 3); (4) IDO-mediated tryptophan degradation was enhanced in the culture medium where the invasion of HTR-8/SVneo trophoblast cells was suppressed.

The mechanism underlying the suppression of IDO-mediated trophoblast cell invasion is unclear at present. It is possible that a decrease in tryptophan concentration induces apoptosis in HTR-8/SVneo trophoblast cells, leading to a reduction in the number of invading cells. Previous studies have suggested that tryptophan depletion, mediated by IDO expressed in decidual macrophages, could contribute to the induction of apoptosis in extravillous cytotrophoblast cells and the regulation of trophoblast invasion [4]. It is also plausible that tryptophan catabolism mediated by IDO may generate metabolites that inhibit trophoblast cell invasion. Quinolinic acid, a downstream metabolite of the IDO-mediated kynurenine pathway, is a potent neuroexcitatory toxin and a mediator of cell destruction in a variety of neurodegenerative disorders [18]. Furthermore, Badawy et al. introduced the concept of tryptophan utilization in pregnancy, proposing that metabolites produced via the kynurenine pathway are essential regulators of immune system activity [19,20].

In early pregnancy, the decidua is densely populated with activated T cells and natural killer cells, which display a unique phenotype distinct from that of peripheral blood natural killer cells [21]. Decidual natural killer cells are recognized for their ability to produce a wide range of cytokines [22], and their secretion of interferon-γ has been well established [23]. IDO is primarily induced by interferon-γ, with additional pro-inflammatory stimuli also contributing to its activation [24]. This suggests that cytokines secreted by these immune cells may play a role in modulating IDO expression. Consequently, the local tryptophan concentration, as determined by the extent of IDO expression, may influence trophoblast invasion at the maternal–fetal interface.

While further studies are needed to elucidate the precise mechanisms by which IDO regulates trophoblast invasion, our findings suggest that IDO expression in the decidua may play a crucial role in controlling extravillous cytotrophoblast invasion at the implantation site, thereby contributing to normal placentation. Additionally, the invasion assay described here should prove useful in the investigation of the molecular mechanisms involved in trophoblast cell invasion. In this assay, molecules potentially involved in invasion are introduced into cells that are initially seeded onto the semi-permeable membrane in the upper compartment of the Transwell insert, enabling the analysis of the molecular basis of trophoblast cell invasion.

## 4. Material and Methods

### 4.1. Sub-Cloning of IDO Gene into Tet-On Inducible Gene Expression Vector System

The human IDO sequence was reverse-transcribed and amplified using term placental RNA with restriction enzyme site-linked primers (IDO-RV-S: GGATATCGACTACAAGAATGGCACACGCTATGG, IDO-Bam-AS: CGGATCCGGATAACCTTCCTTCAAAAGGGATTTCTCAG). PCR fragments were cloned into the pGEM-T easy vector (A1380, Promega, Madison, WI, USA) for sequencing. Clones with 100% sequence identity to IDO (NM_002164.6) were inserted into a modified pFlag-CMV mammalian expression vector (E6908, Sigma Aldrich, St. Louis, MO, USA). Then 1.2 kb size of the IDO fragment was sub-cloned into the multi-cloning site of plasmid vector pTRE2hyg between *SmaI* and *NotI* restriction sites (631070, Takara Bio, Shiga, Japan).

### 4.2. Cell Lines and Stable Transfection

The human trophoblast cell line, HTR-8/SVneo, was kindly provided by Dr. Charles H. Graham, Department of Anatomy & Cell Biology, Queen’s University, Toronto, Canada. The endometrial Ishikawa cell line was generously supplied by Dr Susan Nagel and Dr Danny Schust at The University of Missouri. Cell lines were cultured in DMEM (041-29775, FujiFilm, Tokyo, Japan) supplemented with 10% FBS (Fetal Bovine Serum: Thermo Fisher scientific, Waltham, MA, USA). To develop the doxycycline-responsive derivative of the Ishikawa cell producing the Tet-On transactivator protein, cells were transfected with 1 μg of pTet-On advanced vector (631070, Takara Bio, Shiga, Japan) using 2 μL of lipofectamine-2000 (11668027, Thermo Fisher scientific, Waltham, MA, USA). G418 (400 μg/mL)-resistant clones were selected and were maintained under G418 selection media (100 μg/mL) (A1720, Sigma Aldrich, St. Louis, MO, USA). The G418-resistant Ishikawa Tet-On cells (1 × 10^5^ cells) were subsequently transfected with recombinant pTRE2hyg-IDO plasmid constructs (500 ng) as described above, followed by selection in DMEM media containing 100 μg/mL of G418 and 400 μg/mL of hygromycin (089-06151, FujiFilm, Tokyo, Japan). For isolating a monoclonal IDO expressing Ishikawa cells (Ishikawa Tet-On IDO cell), limited dilution cloning was applied using 96-well plate culturing. To obtain a stable HTR-8/SVneo cell expressing GFP, EGFP fragments from the pEGFP1 vector were cloned into a mammalian expression vector (pCAG vector) and transfected to HTR-8/SVneo cells. HTR-8/SVneo cells expressing GFP (HTR8 GFP) were selected in DMEM 10%FCS media containing 1 μg/mL of puromycin (160-23151, FujiFilm, Tokyo, Japan).

### 4.3. Western Blot

The samples were lysed in RIPA buffer (50 mM Tris-HCl, pH 8.0, 150 mM Sodium Chloride, 0.5 *w*/*v*% sodium deoxycholate, 0.1 *w*/*v*% sodium dodecyl sulfate, 1.0 *w*/*v*% NP-40 substitute with protease inhibitor (165-26021, FujiFilm, Tokyo, Japan)) and analyzed using standard PAGE and Western immunoblotting. NuPAGE Bis-Tris 4–12% precast gels (NW04127, Thermo Fisher Scientific, Waltham, MA, USA) were used for gel electrophoresis according to the instructions. Immunoblotting was performed by using the iBlot2 dry blotting system (IB21001, IB24002: Thermo Fisher scientific, Waltham, MA, USA), and polyclonal anti-IDO antibody (1/3000 dilution; #86630, CST, Danvers, MA, USA) and β-actin antibody (1/5000 dilution; A5441, Sigma Aldrich, MO, USA) were used for detection. The HRP (horseradish peroxidase) signals were detected using a CCD imaging unit (Ez-Capture MG/ST; ATTO corporation, Tokyo, Japan).

### 4.4. Tryptophan Catabolism by IDO

The conditioned medium was collected and centrifuged at 3000× *g* for 5 min and stored at −20 °C until use. The concentrations of tryptophan and kynurenine were measured by the enzyme-linked immunosorbent assay according to the manufacturers’ instructions (ISE-2227R, Immusmol SAS, Bordeaux, France) using 20 μL of supernatants. The ratio of the concentration of kynurenine (a degradation product of tryptophan by IDO) to tryptophan (kynurenine/tryptophan) was used to indicate the functional IDO activity.

### 4.5. Invasion Assay

The in vitro invasive properties of GFP-transfected HTR-8/SVneo trophoblast cells were assessed using a transwell assay system (Figure 4). Ishikawa Tet-On IDO cells (3 × 10^5^ cells) were seeded in the upper compartment of the Transwell insert (662638, Greiner Bio-one, Frickenhausen, Germany) in DMEM without Tet-FCS (631105, Takara Bio, Shiga, Japan) and cultured with or without 200 ng/mL of doxycycline (D5207, Sigma Aldrich, St. Louis, MO, USA). The 200 ng/mL of doxycycline significantly increased both the level of IDO protein expression (Figure 1A) and the degradation of tryptophan (tryptophan concentration: vehicle, 81 ± 6; doxycycline, 74 ± 3 μmol/L; kynurenine concentration: vehicle, 0.81 ± 0.10; doxycycline, 6.26 ± 1.55 μmol/L in the conditioned medium). After 10 h of pre-incubation, HTR8 GFP cells (3 × 10^5^) were layered on Ishikawa Tet-On IDO cells treated with doxycycline or vehicle and incubated for 12 h. The lower chamber was filled with 10% Tet-FCS DMEM medium. The HTR8 GFP cells on the lower surface of the filters which had passed through Ishikawa cells and the semi-permeable membrane were detected and quantified using a fluorescence microscope unit (BZ-X710; KEYENCE Japan, Osaka, Japan). The analysis of the relative invasion ratio was determined using the masking function of the Cell Hybrid counting system software (BZ-X710; KEYENCE Japan, Osaka, Japan). All cells on a semi-permeable membrane were analyzed and the cell invasion ratio was calculated as the relative change compared to vehicle treatment in each assay (i.e., values with vehicle treatment set to 1).

### 4.6. Statistical Analysis

Differences between groups were analyzed using the Mann–Whitney U test and results were considered statistically significant at * *p* < 0.05 and ** *p* < 0.01.

## Figures and Tables

**Figure 1 ijms-26-05889-f001:**
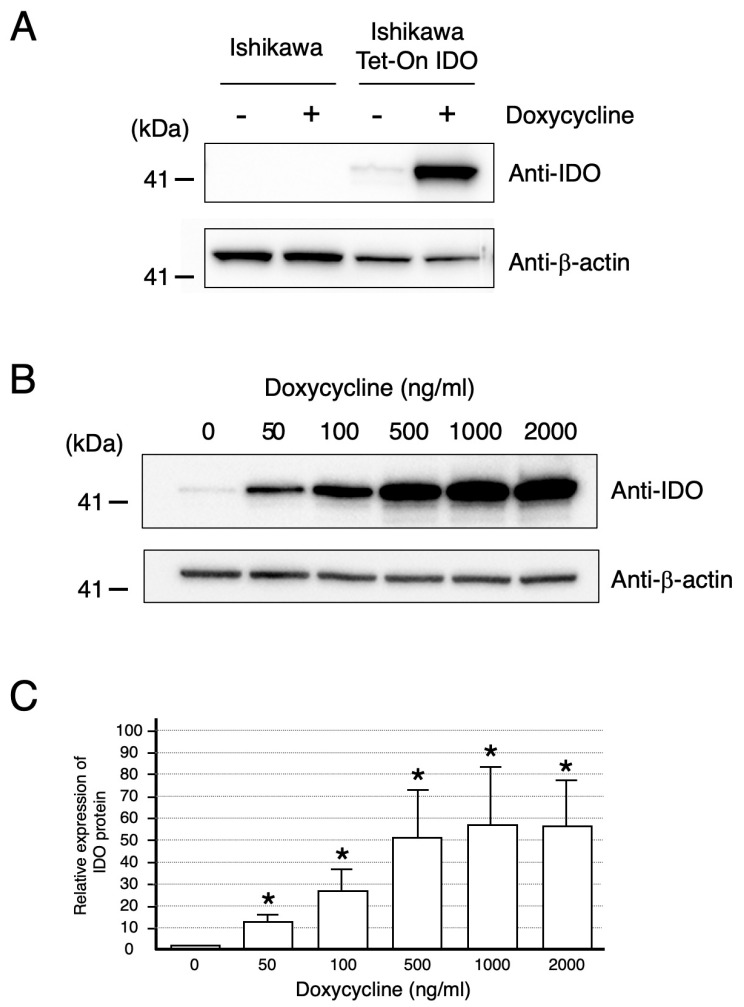
The protein expression level of indoleamine 2,3-dioxygenase (IDO) in Ishikawa cells. (**A**) IDO protein levels in Ishikawa cells and Ishikawa cells stably transfected with a plasmid encoding IDO under the control of a tetracycline inducible promoter (Ishikawa Tet-On IDO cell) treated with 200 ng/mL of doxycycline or vehicle (Nil) for 12 h were analyzed by Western blotting as described in Section 4. The presented results are from a single representative experiment. (**B**) The effect of doxycycline concentration on the stimulation of IDO protein expression level. Ishikawa cells were cultured for 22 h with the indicated concentrations of doxycycline. IDO protein levels were analyzed by Western blotting as described in Section 4. The presented results are from a single representative experiment. (**C**) The relative quantitation of IDO protein level. The intensity of either the IDO or the β-actin band was quantitated by using a gel documentation and analysis system and the ratio of the two was used as a normalized relative abundance value of IDO protein. Data represent the mean ± S.D. of three separate experiments. * Significantly different from values cultured without doxycycline (*p* < 0.05).

**Figure 2 ijms-26-05889-f002:**
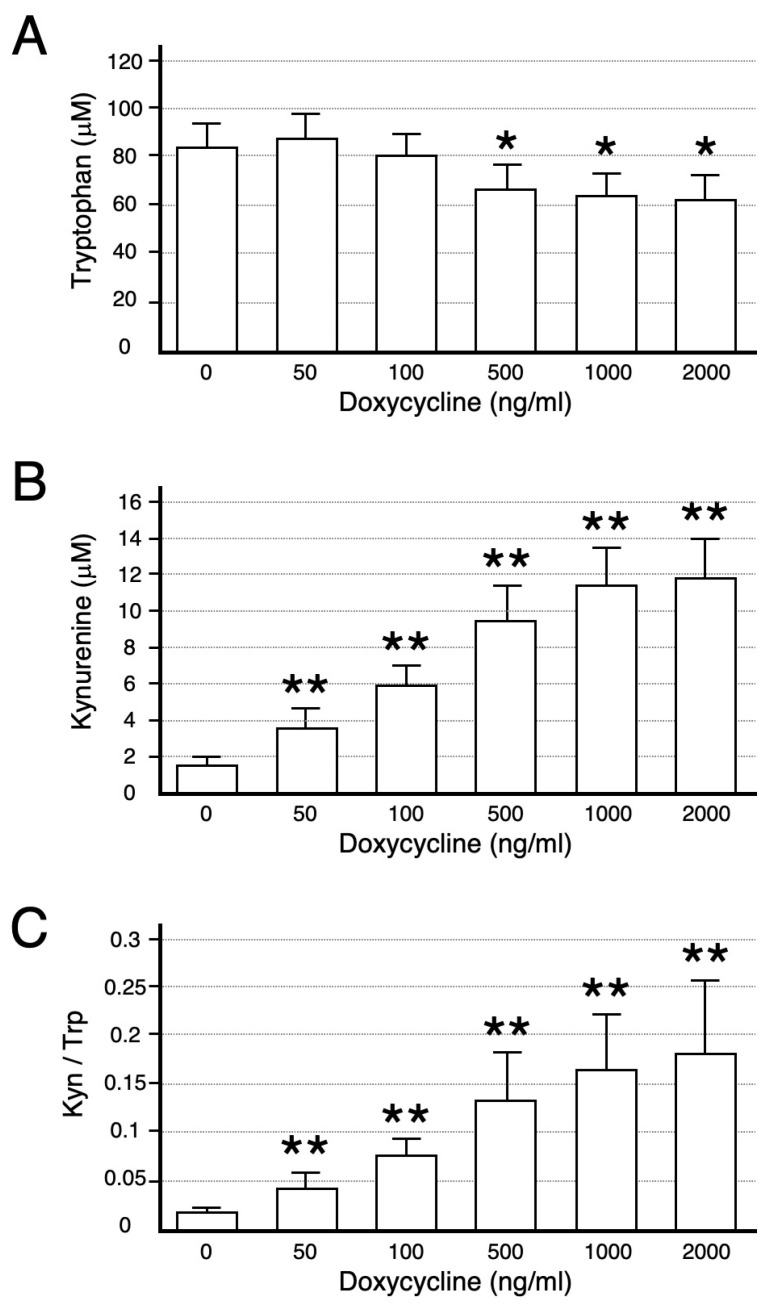
Tryptophan and kynurenine concentrations and the ratio of kynurenine to tryptophan. Concentrations of tryptophan (**A**) and kynurenine (**B**) in the conditioned medium were analyzed as described in Section 4. (**C**) The ratios of kynurenine (Kyn) to tryptophan (Trp). Values are the mean ± S.D. of three separate experiments with a duplicate assay and are significantly different from values cultured without doxycycline (* *p* < 0.05, ** *p* < 0.01).

**Figure 3 ijms-26-05889-f003:**
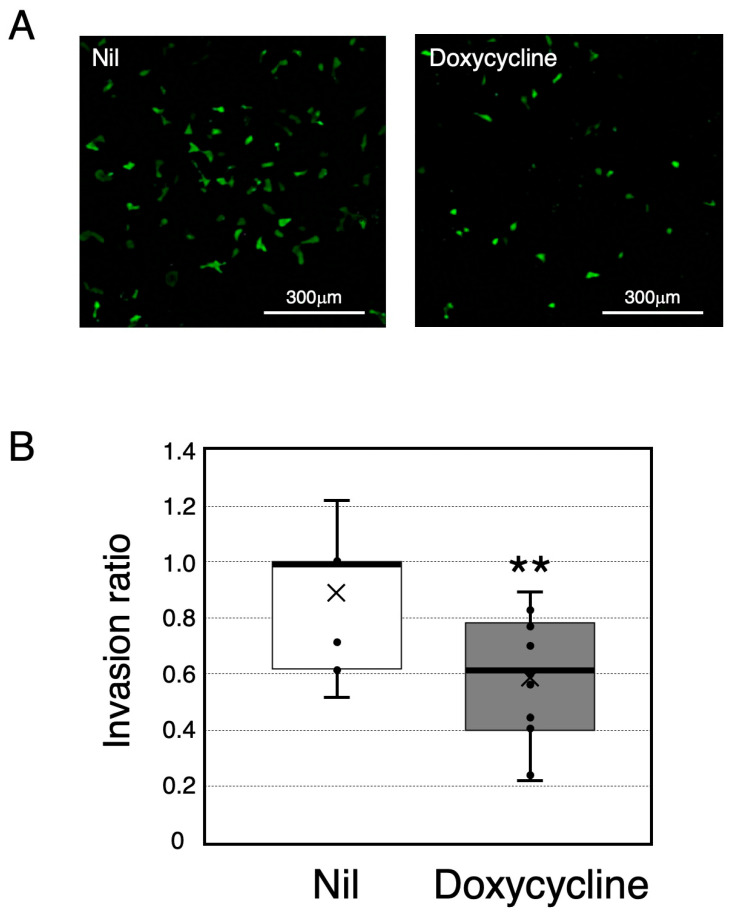
An analysis of HTR-8 cell invasion. HTR-8/SVneo cells stably expressing GFP (HTR8 GFP cell) were cultured on a layer of Ishikawa cells stably transfected with a plasmid encoding IDO under the control of a tetracycline inducible promoter (Ishikawa Tet-On IDO cell) with 200 ng/mL of doxycycline or vehicle (Nil). Then, the number of HTR8 GFP cells which appeared on the opposite side of the permeable membrane was assessed. (**A**) HTR8 GFP cells imaged by fluorescence microscopy. Images were collected by fluorescence microscopy as described in Section 4. Scale bar, 300 µm. (**B**) The number of invaded cells. Data represent box and whisker plots of six separate experiments, expressed as the invasion ratio (i.e., values with vehicle set to 1). ** Significantly different from values cultured without doxycycline (*p* < 0.01).

**Figure 4 ijms-26-05889-f004:**
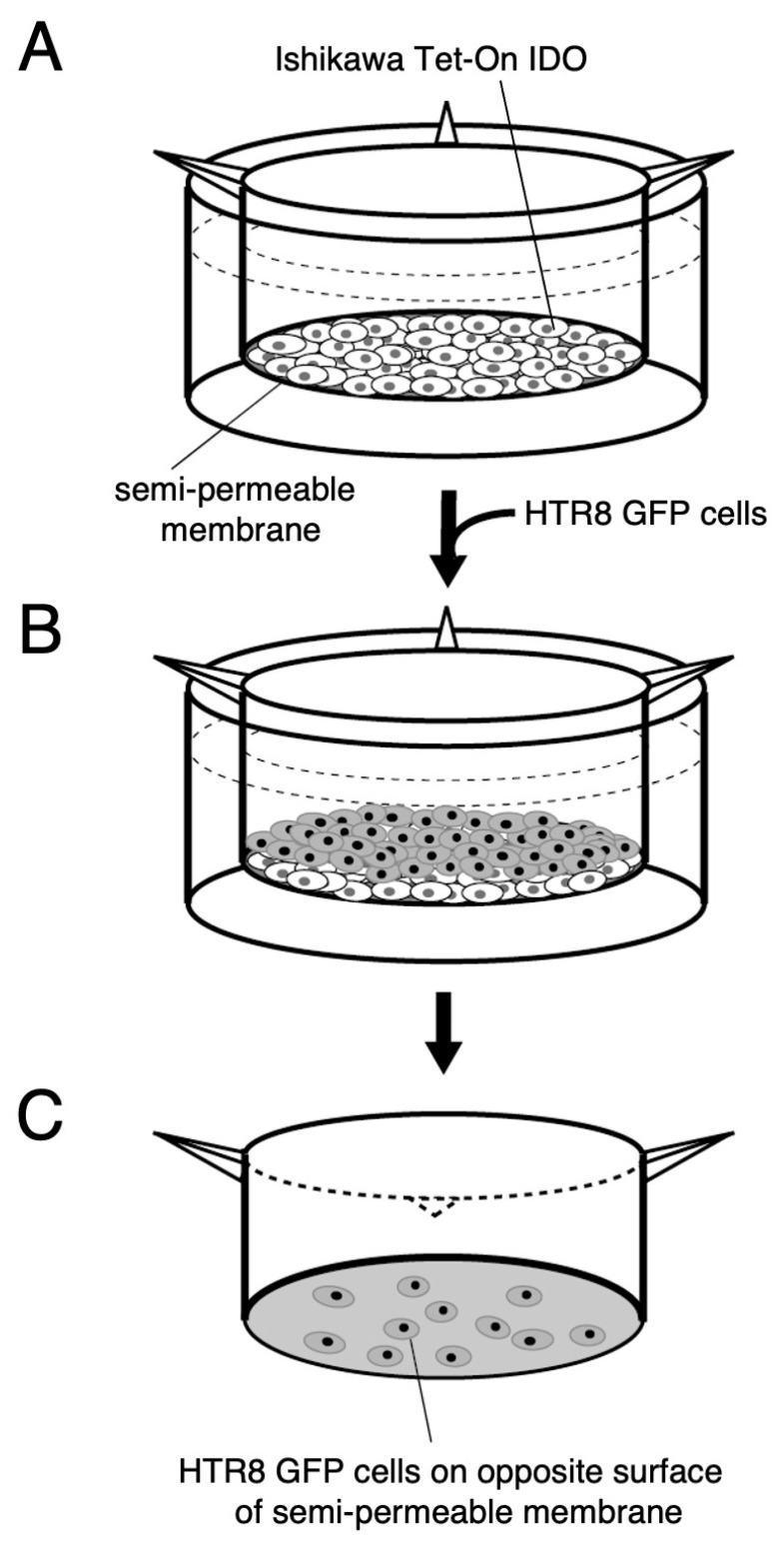
Invasion assay. (**A**) Ishikawa cells stably transfected with a plasmid encoding IDO under the control of a tetracycline inducible promoter (Ishikawa Tet-On IDO cell) were seeded in the upper compartment of the Transwell insert and cultured with 200 ng/mL of doxycycline or vehicle (Nil) in DMEM without Tet-FCS for 10 h. (**B**) HTR-8/SVneo cells expressing GFP (HTR8 GFP) were layered onto Ishikawa Tet-On IDO cells, followed by further incubation for 12 h. The lower chamber was filled with 10% Tet-FCS DMEM medium. (**C**) HTR8 GFP cells which appeared on the opposite surface of the filters which had passed through Ishikawa cells and the semi-permeable membrane were quantified using a fluorescence microscope.

## Data Availability

The original contributions presented in this study are included in the article. Further inquiries can be directed to the corresponding author.

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
