# Peer review of "Indoleamine 2,3-Dioxygenase Regulates Placental Trophoblast Cell Invasion"

_ijms, 2025, doi:10.3390/ijms26125889_

Round 1

Reviewer 1 Report

Comments and Suggestions for Authors

1.The purpose of this study is unclear. From the title "Indooleamine 2,3-dioxygenase mediated trypsin degradation regulations the invasion of human placental trophoblast HTR-8/SVneo cell culture in vitro" and the abstract "To claim the physiological importance of the trypsin catabolizing enzyme, indoleamine 2,3-dioxygenase, in human pregnancy, we have studied how the expression of this enzyme controls extravillous cytotrophoblast invasion into the decision." "These findings suggest that indoleamine 2,3-dioxygenase Dioxygenase Expressed in the Decidua May Play a Role in Regulating Trophoblast Invasion" and Introduction “We then conducted experiments in order to further explore the potential role of IDO in regulating trophoblast invasion”(Line61-62),the research purpose of this paper is to explore the inhibitory effect of Indoleiamine 2,3-dioxygenase mediated trypsin degradation on the invasion of human placental trophoblast HTR-8/SVneo cells. If this is the research objective, the plasmid encoding indoleamine 2,3-dioxygenase can be directly transferred into HTR-8/SVneo cells and its invasive effect can be observed. From the research design and results of this paper, it appears that the purpose of the study is more like to generated Ishikawa cell line stably transfected with a plasmid encoding indoleamine 2,3-dioxygenase under control of a tetracycline inducible promoter, And observe the inhibitory effect of the cell line on the invasion of HTR-8/SVneo cells. This is technical research rather than mechanism research. This is not the same as the goals listed in the title.

2.Line72-73, 139-14: Why is there an underline? What is the purpose?

3.It is recommended to supplement Figure 1 with a bar chart that can display protein expression levels.

4.The expression of numbers in Table 1 is incorrect. Suggest changing "xx * * ± xx" to "xx ± xx * *". Suggest changing Table 1 to a bar chart for better results.

5.Line115-119:“As predicted both the decrease in tryptophan concentration (vehicle, 81  ± 6;  doxycycline, 74 ± 3  μmol/l) and the associated increase in kynurenine concentration (vehicle, 0.81  ± 0.10;  doxycycline, 6.26 ± 1.55  μmol/l)) in the conditioned medium were enhanced significantly by the presence of doxycycline in the culture medium. ”

What method is used to detect the concentration of tryptophan and kynurenine? There is no corresponding introduction in the Materials and Methods section. Please add. If using the "Western Blot" method for detection, how can there be units (μ mol/l)? If using the "Western Blot" method for detection, please provide protein band plots and bar charts. Have the expression levels of trypsin and kynurenine in cells been detected? Please provide additional data.

Author Response

We are very grateful for the careful comments.

1. The purpose of this study is unclear. From the title "Indooleamine 2,3-dioxygenase mediated trypsin degradation regulations the invasion of human placental trophoblast HTR-8/SVneo cell culture in vitro" and the abstract "To claim the physiological importance of the trypsin catabolizing enzyme, indoleamine 2,3-dioxygenase, in human pregnancy, we have studied how the expression of this enzyme controls extravillous cytotrophoblast invasion into the decision." "These findings suggest that indoleamine 2,3-dioxygenase Dioxygenase Expressed in the Decidua May Play a Role in Regulating Trophoblast Invasion" and Introduction “We then conducted experiments in order to further explore the potential role of IDO in regulating trophoblast invasion”(Line61-62),the research purpose of this paper is to explore the inhibitory effect of Indoleiamine 2,3-dioxygenase mediated trypsin degradation on the invasion of human placental trophoblast HTR-8/SVneo cells. If this is the research objective, the plasmid encoding indoleamine 2,3-dioxygenase can be directly transferred into HTR-8/SVneo cells and its invasive effect can be observed. From the research design and results of this paper, it appears that the purpose of the study is more like to generated Ishikawa cell line stably transfected with a plasmid encoding indoleamine 2,3-dioxygenase under control of a tetracycline inducible promoter, And observe the inhibitory effect of the cell line on the invasion of HTR-8/SVneo cells. This is technical research rather than mechanism research. This is not the same as the goals listed in the title.

This study aims to investigate whether indoleamine 2,3-dioxygenase (IDO) expressed in the decidua regulates trophoblast invasion into decidual tissue, using an in vitro experimental model. As described in the Introduction (Line 48-57 in the revised version), we hypothesized that trophoblast invasion into the decidua is regulated by IDO expressed in the decidual tissue, based on our analysis of clinical sample from cesarean scar pregnancy (Journal of Reproductive Immunology 138 (2020) 103099, reference [14]). To investigate this hypothesis using an in vitro model, we established a stable Ishikawa cell line transfected with a plasmid encoding IDO under the control of a tetracycline-inducible promoter. We then examined whether IDO expression suppresses the invasion of HTR-8/SVneo cells into the Ishikawa cell layer. In this in vitro system, HTR-8/SVneo cells were used as a model of trophoblasts, while Ishikawa cells served as a surrogate for endometrial cells.

We agree with the reviewer’s comment regarding the title and have revised it to ‘Indoleamine 2,3-dioxygenase regulates placental trophoblast cell invasion’.

2. Line72-73, 139-14: Why is there an underline? What is the purpose?

Thank you for pointing that out. These were typographical errors and have been corrected.

3. It is recommended to supplement Figure 1 with a bar chart that can display protein expression levels.

As suggested, we have added a bar chart to Figure 1 to present protein expression levels (now shown as Figure 1C in the revised version).

4. The expression of numbers in Table 1 is incorrect. Suggest changing "xx * * ± xx" to "xx ± xx * *". Suggest changing Table 1 to a bar chart for better results.

As suggested, we have changed Table 1 to a bar chart (Figure 2 in the revised version).

5. Line115-119:“As predicted both the decrease in tryptophan concentration (vehicle, 81  ± 6;  doxycycline, 74 ± 3  μmol/l) and the associated increase in kynurenine concentration (vehicle, 0.81  ± 0.10;  doxycycline, 6.26 ± 1.55  μmol/l)) in the conditioned medium were enhanced significantly by the presence of doxycycline in the culture medium. ”

What method is used to detect the concentration of tryptophan and kynurenine? There is no corresponding introduction in the Materials and Methods section. Please add. If using the "Western Blot" method for detection, how can there be units (μ mol/l)? If using the "Western Blot" method for detection, please provide protein band plots and bar charts. Have the expression levels of trypsin and kynurenine in cells been detected? Please provide additional data.

The method used to measure tryptophan and kynurenine levels in the culture supernatant is described in the section ‘4.4. Tryptophan Catabolism by IDO’ of the Materials and Methods. The measurements were performed using an enzyme-linked immunosorbent assay (ISE-227R, Immusmol SAS, France). Intracellular levels of tryptophan and kynurenine were not assessed.

Reviewer 2 Report

Comments and Suggestions for Authors

L59: in vitro probably italic

L186: Clones with 100% sequence identity to IDO: Does this include the untranslated regions?

L, 75, 236: Significant and maximal stimulation were observed at 50 and 1000 ng/ml doxycycline respectively. Cultured with or without 200 ng/ml doxycycline. Could the authors clarify the basis for determining the dose/concentration of doxycycline.

Author Response

We are grateful to the referee for his/her positive comments.

1. L59: in vitro probably italic

We have re-written the text as suggested (page 1, line 14 and page 2, line 58). Thank you!

2. L186: Clones with 100% sequence identity to IDO: Does this include the untranslated regions?

The cloned IDO sequence contains only the open reading frame and does not include any untranslated regions.

3. L, 75, 236: Significant and maximal stimulation were observed at 50 and 1000 ng/ml doxycycline respectively. Cultured with or without 200 ng/ml doxycycline. Could the authors clarify the basis for determining the dose/concentration of doxycycline.

The rationale for using doxycycline at a concentration of 200 ng/ml in the invasion assay (Figure 4 in the revised version) is as follows. As shown in Figures 1B and 1C, doxycycline induced IDO protein expression in Ishikawa Tet-On IDO cells in a dose-dependent manner within the range of 50–1000 ng/ml. Functional IDO activity, assessed by measuring tryptophan and kynurenine levels in the culture supernatant (Figure 2), was consistent with IDO protein expression levels. The selected concentration of 200 ng/mL lies within this effective range and supports both IDO protein induction and enzymatic activity. Furthermore, Figure 1A demonstrates that treatment with 200 ng/ml doxycycline significantly increases IDO protein levels in Ishikawa Tet-On IDO cells. Based on these findings, 200 ng/mL doxycycline was used in the invasion assay shown in Figure 4. As described in the main text (page 4, lines 125–129), doxycycline at this concentration significantly enhanced tryptophan degradation in the culture supernatant and suppressed HTR8 cell invasion.

Round 2

Reviewer 1 Report

Comments and Suggestions for Authors

1.Figure 1-4: It is suggested to replace "A), B), C)" with "A B C".

2. Figure 1-C: The bar chart lacks standard deviation.

3.Line 147-154: It is suggested to replace "1) 2) 3) 4) " with "(1) (2) (3) (4),to avoid confusion with the preceding ().

Author Response

We are grateful for your careful comments. We agree with all of them.

1. Figure 1-4: It is suggested to replace "A), B), C)" with "A B C".

As per your instruction, "A), B), C)" has been replaced with "A, B, C”.

2.Figure 1-C: The bar chart lacks standard deviation.

The standard deviation has been added to Figure 1C, and the figure legend has been revised accordingly. Additionally the original Western blot image corresponding to Figure 1B (C) has been submitted as a supplementary file.

3. Line 147-154: It is suggested to replace "1) 2) 3) 4) " with "(1) (2) (3) (4)” to avoid confusion with the preceding ().

We have re-written the text as suggested. Thank you very much.